



# The UK Environmental Change Network datasets - integrated and co-located data for long-term environmental research (1993-2015)

Susannah Rennie[1], Chris Andrews[2], Sarah Atkinson[3], Deborah Beaumont[4], Sue Benham[5], Vic Bowmaker[6], Jan Dick[2], Bev Dodd[1], Colm McKenna[7], Denise Pallett[8], Rob Rose[1], Stefanie M. Schäfer[8], Tony Scott[9], Carol Taylor[10], Helen Watson[10]

[1]Centre for Ecology & Hydrology, Lancaster Environment Centre, Library Avenue, Bailrigg, Lancaster, LA1 4AP, UK
[2]Centre for Ecology & Hydrology, Bush Estate, Penicuik, Edinburgh, Midlothian, EH26 0QB, UK
[3]Defence Science and Technology Laboratory, Dstl Porton, Porton Down, Salisbury, Wiltshire, SP4 0JQ, UK
[4]Rothamsted Research, North Wyke, Okehampton, Devon, EX20 2SB, UK
[5]Forest Research, CESB, Alice Holt Lodge, Wrecclesham, Farnham, Surrey, GU10 4LH, UK
[6]Cyfoeth Naturiol Cymru - Natural Resources Wales, Maes-y-Ffynnon, Penrhosgarnedd, Bangor, Gwynedd, LL57 2DW, UK
[7]Agri-Food and Biosciences Institute, Newforge Lane, Belfast, BT9 5PX, UK
[8]Centre for Ecology & Hydrology, Maclean Building, Crowmarsh Gifford, Wallingford, Oxon, OX10 8BB, UK
[9]Rothamsted Research, West Common, Harpenden, Herts, AL5 2JQ, UK
[10]The James Hutton Institute, Craigiebuckler, Aberdeen, AB15 8QH, UK

Correspondence to: Susannah Rennie (srennie@ceh.ac.uk)

**Abstract.** Long-term datasets of integrated environmental variables, co-located together, are relatively rare. The UK Environmental Change Network (ECN) was launched in 1992 and provides the UK with its only long-term integrated environmental monitoring and research network for the assessment of the causes and consequences of environmental change. Measurements, covering a wide range of physical, chemical and biological 'driver' and 'response' variables are made in close proximity at ECN terrestrial sites using protocols incorporating standard quality control procedures. This paper describes the datasets (there are nineteen published ECN datasets) for these co-located measurements, containing over twenty years of data (1993-2015). The data and supporting documentation are freely available from the NERC Environmental Information Data Centre under the terms of the Open Government Licence using the following DOI's:

***Meteorology***

Meteorology: https://doi.org/10.5285/fc9bcd1c-e3fc-4c5a-b569-2fe62d40f2f5 (Rennie *et al.*, 2017a)

***Biogeochemistry***

Atmospheric nitrogen chemistry: https://doi.org/10.5285/baf51776-c2d0-4e57-9cd3-30cd6336d9cf (Rennie *et al*., 2017b)

Precipitation chemistry: https://doi.org/10.5285/18b7c387-037d-4949-98bc-e8db5ef4264c (Rennie *et al*., 2017c)

Soil solution chemistry: https://doi.org/10.5285/b330d395-68f2-47f1-8d59-3291dc02923b (Rennie *et al*., 2017d)

Stream water chemistry: https://doi.org/10.5285/fd7ca5ef-460a-463c-ad2b-5ad48bb4e22e (Rennie *et al*., 2017e)

Stream water discharge: https://doi.org/10.5285/8b58c86b-0c2a-4d48-b25a-7a0141859004 (Rennie *et al*., 2017f)

***Invertebrates***

Moths: https://doi.org/10.5285/a2a49f47-49b3-46da-a434-bb22e524c5d2 (Rennie *et al*., 2017g)



Butterflies: https://doi.org/10.5285/5aeda581-b4f2-4e51-b1a6-890b6b3403a3 (Rennie *et al*., 2017h)

Carabid beetle: https://doi.org/10.5285/8385f864-dd41-410f-b248-028f923cb281 (Rennie *et al*., 2017i)

Spittle bugs: https://doi.org/10.5285/aff433be-0869-4393-b765-9e6faad2a12b (Rennie *et al*., 2018)

*Vegetation*

Baseline: https://doi.org/10.5285/a7b49ac1-24f5-406e-ac8f-3d05fb583e3b (Rennie *et al*., 2016a)

Coarse grain: https://doi.org/10.5285/d349babc-329a-4d6e-9eca-92e630e1be3f (Rennie *et al*., 2016b)

Woodland: https://doi.org/10.5285/94aef007-634e-42db-bc52-9aae86adbd33 (Rennie *et al*., 2017j)

Fine grain: https://doi.org/10.5285/b98efec8-6de0-4e0c-85dc-fe4cdf01f086 (Rennie *et al*., 2017k)

*Vertebrates*

Frogs: https://doi.org/10.5285/4d8c7dd9-8248-46ca-b988-c1fc38e51581 (Rennie *et al*., 2017l)

Birds (Breeding bird survey): https://doi.org/10.5285/5886c3ba-1fa5-49c0-8da8-40e69a10d2b5 (Rennie *et al*., 2017m)

Birds (Common bird census): https://doi.org/10.5285/8582a02c-b28c-45d2-afa1-c1e85fba023d (Rennie *et al*., 2017n)

Bats: https://doi.org/10.5285/2588ee91-6cbd-4888-86fc-81858d1bf085 (Rennie *et al*., 2017o)

Rabbits and deer: https://doi.org/10.5285/0be0aed3-f205-4f1f-a65d-84f8cfd8d50f (Rennie *et al*., 2017p)

**1 Introduction**

The assessment of environmental change requires an understanding of how ecosystems function, how they respond to a range of pressures and how resilient they are to such changes. To make these assessments, precise and consistent measurements repeated over long periods of time are needed (Sier and Monteith, 2016a). Ideally, these measurements should also be co-located to provide opportunities to directly link pressures and responses. This type of monitoring effort requires sustained

funding (longer than usual research grants) and a clear long-term vision. Consequently, robust long-term environmental research networks are relatively rare.

The Environmental Change Network (ECN), launched in 1992, is the UK's long-term integrated environmental monitoring and research network (Environmental Change Network, 2019). ECN collects information on a broad baseline of integrated environmental information. The programme also provides more immediate information about trends and early warning of

environmental extremes that may directly influence environmental policy. The ECN programme is sponsored by a consortium of fourteen UK Government departments and agencies (see acknowledgements), who contribute to the programme through funding either site monitoring or network co-ordination activities.

For the period covered by the published datasets, there were twelve terrestrial sites in the network (see figure 1), selected to cover the main range of environmental conditions present in the UK (see table 1). The majority of these sites have been

collecting data since at least 1993, meaning over twenty years of ECN data are now available. However many of the sites were chosen because they had a long history of environmental monitoring so have additional pre-ECN data available.

The monitoring programme includes a wide range of physical, chemical and biological 'driver' and 'response' variables, identified by experts in the field as being important for the assessment of environmental change (see table 2). A Statistical and Technical Advisory Group met regularly to review ECN monitoring activities. These measurements are made in close proximity at each site, using standard protocols incorporating standard quality control procedures (Sykes and Lane, 1996).

Data are managed by the ECN Data Centre, which has an integrated information resource (Rennie, 2016) that stores all data and meta-data collected by the networks which supply data to it. These data are held in standardised structures in order to support the cross-disciplinary analyses necessary for environmental change research. An associated summary database consists of monthly, quarterly, and/or annual summaries of these data using summary statistics appropriate to each measurement, as advised by experts. These summary data can be explored through data visualisation interfaces available on

the website (ECN Data Centre, 2019). The database uses the Oracle relational database management system with links to Arc GIS for spatial data handling. Data were regularly sent in from sites and were quality assured before being lodged in the database (information about quality control is in section 4).

This paper describes the datasets for the high frequency, co-located ECN measurements. There are nineteen published datasets containing over twenty years of data (1993-2015) covering biological, meteorological and biogeochemical measurements

(Rennie *et al*., 2016a,b; Rennie *et al*., 2017a-p; Rennie *et al*., 2018). They are hosted by the NERC Environmental Information Data Centre and are available to users under an Open Government Licence.

## 2. Methods

ECN measurements are co-ordinated and standardised across sites according to published protocols procedures (Sykes and Lane, 1996). The protocol documents are included in the supporting documentation provided alongside every data download.

The protocols are designed to ensure consistency in methods and data handling over time and across ECN's sites. Sites were visited on the same day each week, preferably on a Wednesday, to synchronise sampling, within the site and across the network. The protocol documents detail quality control procedures e.g. correct handling of equipment and samples, maintenance schedules and calibration specifications; as well as unambiguous instructions for measurement and data handling. Data requirements are an integral part of these protocols and include specifications of variables, units, reporting precisions,

dimensions, resolutions, reference systems and quality assurance procedures. These specifications, together with as much information as possible about likely user requirements, were used in design of the database, and the construction of standard formats for data transfer and standard field forms for each dataset. Where available, existing data capture methodologies were used (e.g. the Rothamsted light trap network (Rothamsted Insect Survey, 2019)) to maintain compatibility with other sectoral networks.

At each site, an area of one hectare was selected and permanently marked. This is called the Target Sampling Site (TSS) and destructive sampling within it kept to a minimum. Many of the measurements are co-located within the TSS. Dispersed



monitoring protocols (e.g. vegetation) also include plots within the TSS. The TSS was chosen to be representative of the predominant vegetation, soil and management of the site.

Some protocols (sections 2.15 to 2.19) have not been measured at all sites or have had varied uptake at sites over time, limiting their use for cross-site comparison. In addition, some protocols are designed as national scale surveys so they have limited
use for assessment of trends at individual sites. These limitations are discussed with each individual dataset. The methods for data collection for the nineteen published ECN datasets (1993-2015) are summarised as follows:

### 2.1 Meteorology

Automatic weather stations (AWS) were installed at all ECN terrestrial sites, and situated in accordance with British Meteorological Office site requirements (Meteorological Office, 1982). The AWS was ideally located on, or within 500m of,
the TSS. A number of the sites also had either a manual meteorological station or a second AWS to quality check the data. All ECN AWS instruments were subject to regular (normally annual or biannual) calibration checks. The variables recorded are listed in table 3. Full operating procedures are provided in the protocol document (Burt and Johnson, 1996) which is included in the supporting documentation provided alongside the data download (called MA.pdf).

### 2.2 Atmospheric Nitrogen

Passive diffusion tubes were used to measure the concentration of nitrogen dioxide ($NO_2$) at all ECN terrestrial sites. They were attached to a post at a height of 1.5m above ground level, close to the AWS. As a quality check, blank tubes were also transported to the site but were not exposed on arrival. The blank tubes were returned to the laboratory the same day, stored in a refrigerator and analysed in the lab alongside the experimental tubes. In the early years of ECN, the diffusion tubes were assembled and analysed locally but these were replaced at some sites by commercially made tubes manufactured and analysed
by Gradko Ltd. Comparability tests were conducted when this switch was made. The variables recorded are listed in table 4. Full operating procedures are provided in the protocol document (Bojanic, 1996) which is included in the supporting documentation provided alongside the data download (called AN.pdf).

### 2.3 Precipitation Chemistry

Bulk (open funnel) precipitation collectors were used to measure the precipitation chemistry at all ECN terrestrial sites. These
were situated close to AWS, in an open location away from local sources of contamination (e.g. vehicle tracks or animal houses). Warren Spring Laboratory standard precipitation collectors were used, with the collecting bottle fixed 1.75m above the ground. The collectors were secured by guy ropes or bolted to a concrete base. The collector had a filter to prevent debris falling into the bottle and was kept dark and cool by a jacket. The collecting bottle was changed at the same time each week and the funnel replaced or cleaned with deionised water. The volume collected was recorded, and analysis of the samples were
made by the analytical laboratories linked to each site. The cost of standardising methods of analysis across all ECN laboratories was prohibitive; instead the analytical guidelines (available in supporting documentation available with the data





download) list approved techniques for each determinand with their corresponding limits of detection. Organisations were responsible for maintaining their own continuity in methods for existing long-term runs of data. Each laboratory practised its own internal quality control, and most participated in national quality assurance schemes. As a quality check, a standard quality control solution was sent to the laboratories that analyse the ECN water samples. This solution was analysed alongside

the samples collected in the field. The variables recorded are listed in table 5. Full operating procedures are provided in the protocol document (Adamson and Sykes, 1996) which is included in the supporting documentation provided alongside the data download (called PC.pdf). Operating procedures for handling water samples (Adamson, 1996a) and analytical guidelines (Rowland, 1996) are also provided in the supporting information (called WH.pdf and WAG.pdf).

### 2.4 Soil Solution Chemistry

Water was collected from soils via suction lysimeters at the majority of ECN terrestrial sites. The lysimeters were installed at two depths within a 10m by 10m plot on the edge of, but outside, the TSS. Six samplers were installed in the A horizon and six others at the base of the B horizon (or at 10cm and 50cm if these soil horizons did not exist), ideally on a downslope to avoid debris from soil disturbance. Samplers were emptied and the water volumes collected on the same day each fortnight. One week after sample collection, the samplers were evacuated to 0.5 bar (or 0.7 bar for sites where insufficient soil solution

could be collected), so the water only accumulated over the second week of the fortnightly period. The chemistry of the water collected was analysed by the analytical labs associated with each site. At some sites, particularly in drier months, the volume of water collected may have been very small; in these cases, the samples were discarded or, if possible, combined (only samples from the same horizon were combined) for analysis. The variables recorded are listed in table 5. Full operating procedures are provided in the protocol document (Adamson, 1996b) which is included in the supporting documentation provided

alongside the data download (called SS.pdf). Operating procedures for handling water samples (Adamson, 1996a) and analytical guidelines (Rowland, 1996) are also provided in the supporting information (called WH.pdf and WAG.pdf).

### 2.5 Surface Water Chemistry

Dip samples from rivers and streams were collected. This was only done at sites where flowing water was present. Samples were taken at a representative location above a weir; some sites collect samples at multiple locations on the site (indicated by

the location code in the dataset). The collecting bottle is rinsed in river water and a 250ml sample of river water taken. The variables recorded are listed in table 5. Full operating procedures are provided in the protocol document (Johnson and Burt, 1996a) which is included in the supporting documentation provided alongside the data download (called WC.pdf). Operating procedures for handling water samples (Adamson, 1996a) and analytical guidelines (Rowland, 1996) are also provided in the supporting information (called WH.pdf and WAG.pdf)



### 2.6 Surface Water Discharge

Hydrological data from rivers and streams were collected by logger at sites with a river or stream. Recording of river stage was by a permanently installed weir, the design of which was determined by the conditions at the site. Data were recorded by a logger. The variables recorded are listed in table 2. Full operating procedures are provided in the protocol document (Johnson

and Burt, 1996b) which is included in the supporting documentation provided alongside the data download (called WD.pdf).

### 2.7 Moths

Light traps were used to sample moths (*Macrolepidoptera*) at the majority of the ECN terrestrial sites using the Rothamsted Insect Survey method (Rothamsted Insect Survey, 2019) at the majority of ECN terrestrial sites. Where possible, the light trap was sheltered by vegetation and placed away from artificial light sources, in a location that was convenient for daily emptying.

The traps require a continuous power supply so this often determined their location. Ideally, the traps were emptied daily throughout the year but when this was not possible (e.g. for more remote sites or at the weekend) samples could accumulate. Samples from the sites were identified by a single expert contracted by ECN. The data are stored within the Rothamsted Insect Survey database, as well as in the ECN database. A count of each species trapped was recorded. Full operating procedures are provided in the protocol document (Woiwod, 1996a) which is included in the supporting documentation provided alongside

the data download (called IM.pdf).

### 2.8 Butterflies

Butterfly species were recorded on a fixed transect (which was divided into a maximum of 15 sections) at the majority of ECN terrestrial sites. The transect was chosen to be broadly representative of the site, and include areas under different management regimes. The length of the transect was dependant on the local conditions at the site. The national Butterfly Monitoring

Scheme methodology was used (UK Butterfly Monitoring Scheme, 2019). The transect was walked at an even pace and the number of butterflies which were seen flying within or passing through an imaginary box (5m wide, 5m high and 5m in front of the observer) were recorded. Sampling took place when the temperature was between 13-17°C if sunshine was at least 60%; but if the temperature was above 17°C (15°C at more northerly sites) recording could be carried out in any conditions, providing it was not raining. Transects were walked weekly between the 1[st] April and 29[th] September providing the meteorological

conditions were met. A count of each species observed was recorded. Full operating procedures are provided in the protocol document (Woiwod, 1996b) which is included in the supporting documentation provided alongside the data download (called IB.pdf).

### 2.9 Carabid Beetles

Pitfall traps were used to collect carabid beetles (Carabidae) at the majority of ECN terrestrial sites. Thirty traps were set

divided between three transects, in or adjacent to the TSS; in areas representing different habitats where possible. The traps



were polypropylene measuring 7.5cm diameter by 10cm deep and were filled with ethylene glycol preservative. They were buried with the top of the trap flush with the soil surface. The traps were set 10m apart along the transect. A wire netting cage made from chicken wire, was attached to the rim of the trap to reduce the number of small mammals inadvertently caught. Each trap also had a cover to help prevent rain flooding the traps and reduce bird interference. Samples were analysed by a
local taxonomic expert. The samples were collected fortnightly (between May and end of October). A count of each species trapped was recorded. Full operating procedures are provided in the protocol document (Woiwod and Coulston, 1996) which is included in the supporting documentation provided alongside the data download (called IG.pdf).

### 2.10 Spittle Bugs

Populations of *Philaenus spumaris* and *Neophilaenus lineatus* were monitored annually at the majority of ECN terrestrial sites.
In mid-June, counts of the spittle produced by nymphs made in 20 quadrats ($0.25m^2$) randomly placed near the TSS. Also, in late August, the proportions of each colour morph of the adult *P. spumaris* were estimated using sweep netting on the TSS when the weather conditions were dry. Colour polymorphism is likely to be environmentally determined (Whittaker, 1965) and therefore an indicator of environmental change. The samples were collected annually (nymphs in June and adults in August). A count of each species/colour morph was recorded. Full operating procedures are provided in the protocol document
(Whittaker, 1996) which is included in the supporting documentation provided alongside the data download (called IS.pdf).

### 2.11 Baseline Vegetation

This was a one-off survey at the start of ECN monitoring to establish a vegetation map at all sites. It allowed a vegetation map to be generated and the plots for continuous monitoring (see 2.12, 2.13, 2.14)) to be selected. An approximately regular grid, coincident with the UK National Grid, was superimposed on the site map, scaled so as to provide approximately 400 sample
grid positions. This ensured the plot locations were unbiased and relocatable. Additionally, no more than 100 points (infill points) were chosen to ensure all vegetation types were represented. A 2m x 2m plot was centred on each grid and infill point, oriented using magnetic bearings. These plots were permanently marked (the plot corners are marked with buried metal stakes). Species presence was recorded in the plots. Where the plots fell in woodland, the trees and shrubs were recorded in a 10m x 10m plot centred on the 2m x 2m plot to provide a more representative sample of the canopy and understory. Full
operating procedures are provided in the protocol document (Rodwell *et al.*, 1996) which is included in the supporting documentation provided alongside the data download (called V.pdf).

### 2.12 Coarse-grain Vegetation

2m x 2m plots were randomly selected (from the plots selected in the Baseline Survey) at the majority of ECN terrestrial sites at the onset of ECN monitoring. The total number of plots selected varied in proportion to the total area of the site. Where
plots fell in woodland or scrub, the associated woodland protocol was also undertaken (see 2.13). The protocol was undertaken every nine years. Species presence was recorded in each of the twenty-five 40cm x 40cm cells within the plots. Full operating



procedures are provided in the protocol document (Rodwell *et al.*, 1996) which is included in the supporting documentation provided alongside the data download (called V.pdf).

### 2.13 Woodland Vegetation

10m x 10m plots (which were centred on the 2m x 2m plot used in the coarse-grain survey) were used to record trees and

shrubs. Species dominance was assessed within the plots. Ten cells, each 40cm x 40cm, were selected at random within the plot and marked. Seedlings were counted by species in each cell. Additionally, an individual tree was chosen nearest the centre point of the cell and monitored for height and diameter at breast height (dbh). The protocol was undertaken every nine years, but dbh was measured every three years for sites where there was woodland. The variables recorded are listed in table 6. Full operating procedures are provided in the protocol document (Rodwell *et al.*, 1996) which is included in the supporting

documentation provided alongside the data download (called V.pdf).

### 2.14 Fine-grain Vegetation

10m x 10m plots were randomly selected (from the plots selected in the Baseline Survey) within each vegetation type present on the majority of ECN sites. Ten 40cm x 40cm cells were selected randomly within these plots. This survey was undertaken every three years but some sites chose to do this survey annually to provide a better temporal range. Often they chose a smaller

number of plots to do the annual survey. Species presence was recorded within the cells. Full operating procedures are provided in the protocol document (Rodwell *et al.*, 1996) which is included in the supporting documentation provided alongside the data download (called V.pdf).

### 2.15 Frogs

It is difficult to monitor populations of adult frogs, therefore phenological observations were made of in selected pools and

ditches and the number of egg masses were assessed as an indicator of the 'health' of frog populations at sites with standing water present. Additionally, a 250ml water sample was taken from the spawning area and analysed. The variables recorded are listed in table 7. Full operating procedures are provided in the protocol document (Beattie *et al.*, 1996) which is included in the supporting documentation provided alongside the data download (called BF.pdf).

### 2.16 Birds - Breeding Bird Survey

Bird species were recorded on two transect lines (within a 1km square) at the majority of ECN sites. Counts were made in the morning, ideally no later than 09:00. Transects were walked, at a slow and methodical pace, when the visibility was good and there was no strong wind or heavy rain. All birds seen or heard were recorded, as well as their distance (there are four distance categories) from the transect. The methodology used was that of the Breeding Birds Survey (BBS, 2019) organised by the British Trust for Ornithology (BTO). The transect was walked twice each year (once between April and mid-May and the



second between mid-May to late June). Full operating procedures are provided in the protocol document (Sykes, 1996a) which is included in the supporting documentation provided alongside the data download (called BB.pdf).

This protocol replaced the Common Bird Census (see 2.17) in 1999. Please note that the Breeding Birds Survey is designed to be a national-scale survey. Therefore the site-based ECN data is limited in the amount of information which it can provide

on the precise relationships between population levels and environmental change. It is recommended that the ECN data are used in conjunction with data from more widespread monitoring programmes (i.e. those of the BTO) so these limitations can be mitigated.

### 2.17 Birds - Common Bird Census

Bird species were recording in a plot which should ideally be a minimum of 40 hectares in farmland and 10 hectares in

woodland. The methodology used was that of the Common Birds Census (CBC, 2019) organised by the BTO. Ten visits were made between mid-March and late June, spaced evenly through the season. Cold, windy and wet days were avoided. The CBC uses a mapping method in which a series of visits were made to all parts of a defined plot during the breeding season and contacts with birds by sight or sound were recorded on large-scale maps. Information from the series of visits was combined to estimate the number of territories found. Within the CBC protocol, some species were also monitored by nest counts on the

plot, or by a combination of nest counts and territory estimation. Full operating procedures are provided in the protocol document (Sykes, 1996b) which is included in the supporting documentation provided alongside the data download (called BC.pdf).

The CBC was the standard protocol at lowland ECN sites until 1999 when it was replaced by the BBS (see 2.16). A few sites continued the CBC alongside the BBS for a few years to allow comparison. Additionally, historical data (pre-ECN) was

obtained for the Wytham site. Therefore the date ranges for individual sites in this dataset are not consistent. As with the BBS, the CBC was designed to be a national-scale survey so similar limitations apply to the site-based ECN data provided in this dataset.

### 2.18 Bats

Bat species were mapped (using a bat detector) and their behaviour recorded at the majority of ECN sites. One or more

kilometre squares were selected at the site. This selection did not need to be random as long as the square was reasonably typical of the site and that fieldwork at night could be conducted safely. The square was divided into two and a transect selected through each of these half-squares. The methodology was based on that used in the Bats and Habitats survey organised for the Joint Nature Conservation Committee. The transect was walked four times in each year (once in each three-week period between June and September). Bat detectors were used during the survey and the frequency the detector was tuned to

could be altered during the survey if that helped ensure all species were recorded (in particular to distinguish between Pipistrelle species). Surveys were not carried out when rain was heavy or there were strong winds. A count of each species observed

and their behaviour was recorded. Full operating procedures are provided in the protocol document (Walsh *et al.*, 1996) which is included in the supporting documentation provided alongside the data download (called BA.pdf).

The methodology is somewhat limited in the amount of information which it can provide on the precise relationships between population levels and environmental change; nevertheless by linking ECN results to those from more widespread monitoring

programmes these limitations can be mitigated.

**2.19 Rabbits and Deer**

There were no practicable methods of making direct measures of the population size of the rabbit and deer populations; therefore an index method based on dropping counts was used to estimate relative abundance at the majority of ECN sites. The butterfly monitoring transect was used. A second transect that covered habitat types not present on the butterfly transect

was also selected. Dropping counts were recorded on a transect twice a year (once in late March and again in late September). Droppings on the transect were cleared two weeks before sampling took place. At Moor House, the same methodology was also used to estimate the relative abundance of Grouse. Full operating procedures are provided in the protocol document (Coulson, 1996) which is included in the supporting documentation provided alongside the data download (called BU.pdf).

**3. Datasets**

The ECN datasets are listed in table 2, together with their citation information, the frequency of measurement and the variables collected. Each dataset follows the same basic structure:

- SITECODE - site code (see table 1)
- SDATE - date of sampling
- FIELDNAME - the variable being measured (these are described below and in the supporting information)
- VALUE - the value of the measured variable

All the datasets have this structure in common but some of the datasets may also contain some additional information, where necessary for the measurement. This is fully documented in the supporting information.

The supporting information, i.e. the protocol document, supplementary data and quality information, is provided with each dataset. It is important to refer to this information prior to analysing the data. The NERC Environmental Information Data

Centre (the repository that hosts the datasets) provides data and supporting information as separate packages – this allows improvements to be made to the supporting documentation over time if necessary while maintaining a persistent, citable dataset. The DOI for each dataset links to a landing page which contains separate links to download the data and the supporting information. The supporting information is provided in a zip file using the 'Supporting documentation' link on the relevant page for each dataset (Rennie *et al.*, 2016a,b; Rennie *et al.*, 2017a-p; Rennie *et al.*, 2018). All the zip files contain a document

called ***_DATA_STRUCTURE.doc (where *** is the ECN measurement code (see table 2)). This document contains





detailed information about the structure of the dataset, location information for the sites, information about the variables measured, documents for any additional information needed to understand the dataset and provides any coding lists used. Some usage notes are included below:

### 3.1 Meteorology

Given the size of this dataset, the data have been split into yearly csv files. Users are advised to open these files in a text editor or use statistical package to analyse these data as the file sizes remain large.

Over the period of data collection, the majority of ECN sites have operated more than one AWS in the same location – e.g. when kit is replaced. In many cases, these have been run concurrently to enable cross-checking of data. Replacement AWS's are indicated by the 'AWSNO' field in the dataset. Users should be aware of the AWSNO when analysing the data –

particularly when two AWS's have been run concurrently – to avoid misleading results by inadvertently combining data from two AWS'.

### 3.2 Soil Solution Chemistry

Where samples were combined, this is indicated in the data with the replicate IDs XXS (combined shallow samplers) and XXD (combined deep samplers) in the datasets. Occasionally, the suction samplers were replaced, this is indicated in the data with

a new replicate ID.

### 3.3 Surface Water Discharge

Given the size of this dataset, the data have been split into yearly csv files.

One site (Moor House – Upper Teesdale) uses an Environment Agency logger to record water discharge. The Environment Agency uses the WISKI format to record these data (the Hydrolog format was used prior to 2004). Both these formats include

quality information which are available in this dataset (for Moor House only). An explanation for these quality codes is provided in the supporting information.

### 3.9 Carabid Beetles

There is an additional data column in this dataset which applies to only one species (*Pterostichus madidus*) where additional information was collected on gender (M or F) and leg colour (R (red) and B (black)). The ratio of leg colour is thought to

depend on ecological factors.

### 3.5 Species coding lists

ECN uses a number of coding lists within its datasets. Where possible, existing coding systems were used to maintain compatibility with other data resources. These coding lists are fully documented in the supporting information.



### 3.6 Dataset Completeness

The majority of the ECN sites have been collecting the full suite of ECN measurements since 1993 but two sites joined the network later – Yr Wyddfa (Snowdon) in 1995 and Cairngorms in 1999. However, it should be noted that many of the sites are in remote locations which can mean that site managers are unable to attend the sites occasionally for health and safety

reasons, causing gaps in the dataset. In particular, there was a Foot-and-mouth disease outbreak in the UK in 2001, which meant a number of the sites could not be visited for biosecurity reasons, and the data for that year are patchy. In addition, Rothamsted ceased biological monitoring in 2011 and Drayton left the network in 2014.

### 4 Data Quality

Quality control is central to all stages of ECN data collection and management and is handled through:

### 4.1 Standard Operating Procedures.

As described in the section 2, data collection procedures were co-ordinated and standardised across the sites through published protocols.

### 4.2 Data Transfer Templates

Data were checked and formatted by data providers prior to being submitted by email (in standardised, comma-separated files).

Detailed data transfer documentation for each protocol guided the preparation of these files, to ensure comparability of data across sites and over time. This documentation includes rules for handling missing values and data quality information. To aid site managers a bespoke set of data entry templates were developed, using MS Access, to improve data handling efficiency. These templates incorporate quality checking procedures, and help to ensure that quality-checked, standardised and formatted data were submitted by site managers. The design of the templates takes into account ease of use, with the main emphasis

being on minimising error. This type of data entry software is particularly useful where numeric coding systems for species are in use; numbers are less memorable and mistakes in one digit of a code can produce serious errors. For example, the software uses drop-down lists of codes (which are dynamically linked with a list of the species names) so that the codes can be cross-checked against the species name to ensure that the correct code is chosen.

### 4.3 Data Verification

Standard verification procedures were applied to all data before import into the database. The procedures performed numeric range checks (i.e. checking if a value falls within a specified range), categorical checks (e.g. checking that a species code appears on the standard code list), formatting (i.e. that the dataset conforms to the specified data format) and logical integrity checks (i.e. checking the data make sense e.g. that the dates in one dataset match those in a related dataset). Appropriate range



settings for ECN variables were selected following discussion with specialists in each field. Where values fell outside these ranges, a cautious approach was adopted towards discarding data on the principle that apparent errors could be valid outliers. Data values identified by validation software as 'out of range' were treated in one of three ways:

- where values were clearly meaningless due to a known cause, (e.g. an instrumentation fault, and could not be back-corrected), the data were discarded and database fields set to null (no data), and quality flags added to the database;
- where values were clearly in error, or out of range due to known calibration errors and could be back-corrected, the data were corrected. These changes were flagged in the database;
- where there was no straightforward explanation for outliers, the data were stored in the database, accompanied by quality flags.

**4.4 Quality Flagging**

The ECN site managers assigned quality codes to indicate factors that may affect the quality of the data being collected, including deviations from the protocol, faulty instrumentation and common problems. They picked these from a standard list of ECN quality codes and these quality codes are included in the data download and an explanation for the codes provided in the supporting documentation. Site managers could pick as many quality codes as were applicable. Occasionally, an unusual
event took place that was not covered by these codes. In that case, the site manager attached text explaining the circumstances. This is indicated by a quality code '999' in the data download. This quality text is available in a file called ECN_***_qtext.csv (where *** is the measurement code; see table 2) which is provided in the supporting documentation.

**4.5 Quality Assessment Exercises**

Samples were kept where possible (e.g. archived invertebrate samples) meaning the accuracy of identification can been
assessed at a later date if necessary. Occasionally, quality assessment exercises have been run by appropriate experts to check, for example, consistency in species identification across sites (Scott and Hallam, 2003). The quality of more ephemeral measurements such as meteorology or water quality can only be similarly assessed by running duplicate or parallel systems. Duplicate systems are expensive, and in practice assessment normally involved regular checks for instrument drift and recorder error. Where possible, when new instrumentation or methods needed to be introduced, new and old systems were run in
parallel to assess their relationship.

**5. ECN datasets in context**

ECN is nationally unique with its focus on high frequency and co-located measurements. It provides a rare opportunity to link pressures and responses to investigate relationships between environmental variables and explore environmental change over significant time scales. The data included within these datasets have been the focus of a number of peer-reviewed scientific
publications over the past 20 years. For example, linking meteorological with invertebrate species data to explore the impact of drought (Morecroft *et al*., 2002); exploring trends in the physical and biological environment (Morecroft *et al.*, 2009);





determining that hydrochloric acid deposition was a driver of UK soil acidification (Evans, *et al.*, 2011); and to investigate declines in carabid beetle biodiversity (Brooks *et al.*, 2012). Many of the datasets were incorporated in papers forming a journal special issue marking the first 20 years of ECN (Sier and Monteith (*eds.*), 2016b). This special issue demonstrates how effective the datasets are in assessing and interpreting environmental change, covering a breadth of topics such as trends

in weather and atmospheric deposition (Monteith, *et al.*, 2016); trends in dissolved organic carbon (Sawicka *et al.*, 2016; Moody *et al.*, 2016); various aspect of change in UK plant communities (Rose *et al.*, 2016; Morecroft *et al.*, 2016; Pallett *et al.*, 2016; Milligan *et al.*, 2016); ecosystem services (Dick *et al.*, 2016); carabid beetle communities (Eyre *et al.*, 2016; Pozsgai *et al.*, 2016); the use of digital imaging to assess vegetation cover (Baxendale *et al.*, 2016); and the response of Lepidoptera communities to warming (Martay *et al.*, 2016). A full catalogue of the peer-reviewed papers that have used ECN data are

available on the website (ECN Publications Catalogue, 2019).

ECN sites cover a wide range of UK habitats but, given their focus on high frequency data, are costly to run and are relatively few in number. The representativeness of ECN sites was compared to data obtained by the UK Countryside Survey (CS - Countryside Survey, 2019). The survey is based on a stratified random sample of 1 km squares from the intersections of a regular 15 km grid superimposed on the rural areas of Great Britain. Analysis revealed that ECN sites effectively span the

range of values for both temperature and rainfall and cover a similar range of vegetation types to the CS with the exception of arable, a land use category not assessed for vegetation at ECN sites although present on several sites (Dick *et al.*, 2011).

ECN sites contribute to a number of national monitoring programmes e.g. Rothamsted Insect Survey (Rothamsted Insect Survey, 2019), Countryside Survey (Countryside Survey, 2019), the UK Butterfly Monitoring Scheme (UKBMS, 2019), the Breeding Bird Survey (BBS, 2019), the United Kingdom Eutrophying and Acidifying Network (UKEAP, 2019) and the

Cosmic-ray Soil Moisture Monitoring Network (COSMOS-UK, 2019). ECN's focus on multidisciplinary, co-located measurements can help integrate these networks and provides temporal scale context for observations made by these networks, for example by providing information on year to year variation in vegetation communities to help inform how CS data can be influenced by weather variability (Scott *et al.*, 2010).

ECN is formally recognised as the UK's contribution to a global system of long-term, integrated environmental research

networks and is a member of LTER-Europe (the European Long-Term Ecosystem Research Network – Mirtl, 2010) and ILTER (International Long Term Ecological Research – Kim, 2006). Individual ECN sites are also involved in other international networks, including INTERACT (International Network for Terrestrial Research and Monitoring in the Arctic - INTERACT, 2019), GLORIA (Global Observation Research Initiative in Alpine Environments - GLORIA, 2019), ICP Forest Level II (ICP Forests, 2019) and FLUXNET (FLUXNET, 2019).

**6. Data Availability**

Provision of easy access to data has always been central to ECN's strategy to provide a resource for environmental research, policy purposes and public information. The ECN datasets are hosted by the NERC Environmental Information Data Centre



(EIDC, 2019) managed by the Centre for Ecology and Hydrology (CEH). The EIDC manages nationally important terrestrial and freshwater science datasets. The ECN datasets can be discovered and downloaded through the EIDC's data catalogue (the Environmental Information Platform (EIP)). The datasets are listed in table 2, together with their citation information. They should be cited for every use using the information provided (Rennie *et al*., 2016a,b; Rennie *et al*., 2017a-p; Rennie *et al*., 2018).

The ECN datasets are available under the Open Government Licence (Open Government Licence, 2019) and they are available as comma-separated files. Temporal extensions to the datasets will be created as additional years of data become available.

## 7. Conclusions

The datasets collected by the UK Environmental Change Network are an invaluable and nationally unique resource, which, over the years, has proved useful to a range of users, including the scientific community and national policy makers. The co-location of high frequency meteorological, biological and biogeochemical measurements means the ECN datasets are ideally placed for the development of clearer process understanding and assessing the impact of shorter term events, such as droughts, on ecosystems. This two decade ECN data record provides a long-term baseline of environmental variability across a wide range of UK habitats against which environmental changes can be assessed.

## Team List

The following people were ECN site managers during the period of data collection of these datasets: John Adamson, Roy Anderson, Chris Andrews, Sarah Atkinson, John Bater, Neil Bayfield, Clive Bealey, Katy Beaton, Deb Beaumont, Sue Benham, Vic Bowmaker, Chris Britt, Rob Brooker, David Brooks, Andrew Brunt, Jacqui Brunt, Sam Clawson, Gordon Common, Richard Cooper, Stuart Corbett, Nigel Critchley, Peter Dennis, Jan Dick, Bev Dodd, Nikki Dodd, Neil Donovan, Jonathan Easter, Edward Eaton, Mel Flexen, Andy Gardiner, Dave Hamilton, Paul Hargreaves, Maggie Hatton-Ellis, Mark Howe, Olly Howells, Jana Kahl, Simon Langan, Dylan Lloyd, Mathieu Lundy, Briege McCarney, Yvonne McElarney, Colm McKenna, Simon McMillan, Frank Milne, Linda Milne, Mike Morecroft, Matt Murphy, Allison Nelson, Harry Nicholson, Denise Pallett, Dafydd Parry, Imogen Pearce, Gabor Pozsgai, Adrian Riley, Rob Rose, Stefanie Schäfer, Tony Scott, Chris Shortall, Phil Smith, Roger Smith, Richard Tait, Carol Taylor, Michele Taylor, Maddie Thurlow, Christine Tilbury, Alex Turner, Ken Tyson, Helen Watson, Mike Whittaker, Matthew Wilkinson, Ian Woiwod and Christopher Wood.

ECN and its Data Centre are co-ordinated by the Central Co-ordination Unit at CEH Lancaster. The following people have been involved in this during the period of data collection of these datasets: John Adamson, Chris Benefield, Deirdre Caffrey,



Bill Heal, Pete Henrys, Lynne Irvine, Mandy Lane, Don Monteith, Mike Morecroft, Terry Parr, Susannah Rennie, Rob Rose, Andy Scott, Lorna Sherrin, Andy Sier, Ian Simpson and Mike Sykes.

**Author Contributions**

S.R. was responsible for the management of the ECN Data Centre, publication of the datasets and led the writing of this manuscript. C.A., S.A, D.B, S.B, V.B., J.D., B.D., C.M., D.P., R.R., S.S., T.S., C.T., H.W. are the current site managers and are responsible for site management, data collection and quality checking. All co-authors contributed to the writing, discussion and review of this manuscript.

**Competing Interests**

The authors declare that they have no conflict of interest.

**Acknowledgements**

Central co-ordination of ECN is funded by NERC, through CEH (NEC06397). The ECN programme is sponsored by a consortium of UK government departments and agencies who contribute to the programme through funding either site monitoring or network co-ordination activities: Agri-Food and Biosciences Institute, Biotechnology and Biological Sciences Research Council, Cyfoeth Naturiol Cymru - Natural Resources Wales, Defence Science & Technology Laboratory, Department for Environment, Food and Rural Affairs, Environment Agency, Forestry Commission, Llywodraeth Cymru - Welsh Government, Natural England, Natural Environment Research Council, Northern Ireland Environment Agency, Scottish Environment Protection Agency, Scottish Government, Scottish Natural Heritage. The ECN work carried out at Rothamsted and North Wyke forms part of the LTE National Capability programme (BBS/E/C/000J0300) funded by the Biotechnology and Biological Sciences Research Council.

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





**Figure 1: Locations of the ECN terrestrial sites**

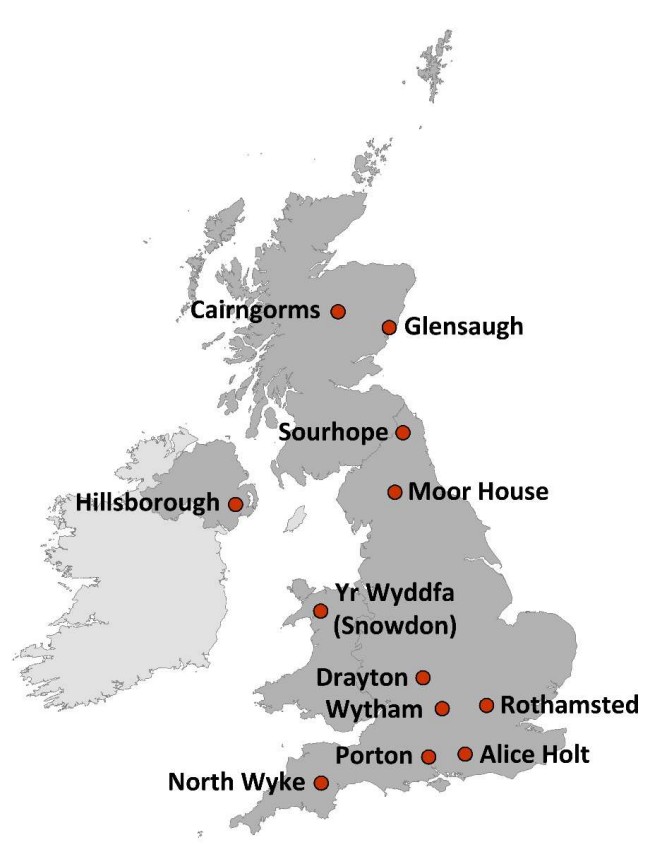

5    **Table 1: ECN terrestrial sites**

| Site (ECN Site code) | Location | Altitudinal Range (m above sea level) | Area (ha) | Site Type |
|---|---|---|---|---|
| Alice Holt (T09) | 51° 9'16.46"N 0°51'47.58"W | 110-125 | 850 | Woodland |
| Drayton (T01) | 52°11'37.95"N | 320-1110 | 1000 | Lowland grassland/agricultural |





| | 1°45'51.95"W | | | (data collection ceased at this site at the end of 2013) |
|---|---|---|---|---|
| Cairngorms (T12) | 57° 6'58.84"N 3°49'46.98"W | 40-80 | 190 | Upland moor/mountain |
| Glensaugh (T02) | 56°54'33.36"N 2°33'12.14"W | 137-487 | 1125 | Upland moor/mountain with native mixed pine wood |
| Hillsborough (T03) | 54°27'12.24"N 6° 4'41.26"W | 110-170 | 400 | Lowland grassland/agricultural |
| Moor House – Upper Teesdale (T04) | 54°41'42.15"N 2°23'16.26"W | 290-848 | 7500 | Upland moor/mountain |
| North Wyke (T05) | 50°46'54.96"N 3°55'4.10"W | 120-180/ | 250 | Lowland grassland/agricultural |
| Porton Down (T10) | 51° 7'37.83"N 1°38'23.46"W | 100-172 | 1227 | Lowland grassland |
| Rothamsted (T06) | 51°48'12.33"N 0°22'21.66"W | 94-134 | 330 | Lowland grassland/agricultural |
| Sourhope (T07) | 55°29'23.47"N 2°12'43.32"W | 200-601 | 1119 | Upland moor/mountain |
| Wytham (T08) | 51°46'52.86"N 1°20'9.81"W | 60-165 | 770 | Woodland/agricultural |
| Yr Wyddfa (Snowdon) (T11) | 53° 4'28.38"N 4° 2'0.64"W | 298-1085 | 700 | Upland moor/mountain |

**Table 2: ECN Datasets**

| Measurement (ECN measurement code) | Frequency of data collection | Variable/s recorded | DOI (Citation) |
|---|---|---|---|
| Meteorology (MA) | Hourly summaries calculated from 5 second samplings | See table 3 | https://doi.org/10.5285/fc9bcd1c-e3fc-4c5a-b569-2fe62d40f2f5 (Rennie *et al*., 2017a) |
| Atmospheric nitrogen (AN) | Fortnightly | See table 4 | https://doi.org/10.5285/baf51776-c2d0-4e57-9cd3-30cd6336d9cf (Rennie *et al*., 2017b) |



| Precipitation chemistry (PC) | Weekly | See table 5 | https://doi.org/10.5285/18b7c387-037d-4949-98bc-e8db5ef4264c (Rennie *et al.*, 2017c) |
|---|---|---|---|
| Soil solution (SS) | Fortnightly | See table 5 | https://doi.org/10.5285/b330d395-68f2-47f1-8d59-3291dc02923b (Rennie *et al.*, 2017d) |
| Surface water chemistry (WC) | Weekly | See table 5 | https://doi.org/10.5285/fd7ca5ef-460a-463c-ad2b-5ad48bb4e22e (Rennie *et al.*, 2017e) |
| Surface water discharge (WD) | 15-minute averages calculated from ten second samplings of stage height | • Stage (m)<br>• Discharge (cumecs) | https://doi.org/10.5285/8b58c86b-0c2a-4d48-b25a-7a0141859004 (Rennie *et al.*, 2017f) |
| Moth (IM) | Nightly; weekly at remote sites | Count of each species trapped | https://doi.org/10.5285/a2a49f47-49b3-46da-a434-bb22e524c5d2 (Rennie *et al.*, 2017g) |
| Butterfly (IB) | Weekly between April and September – dependant on weather conditions | Count of each species observed | https://doi.org/10.5285/5aeda581-b4f2-4e51-b1a6-890b6b3403a3 (Rennie *et al.*, 2017h) |
| Carabid beetles (IG) | Fortnightly | Count of each species trapped | https://doi.org/10.5285/8385f864-dd41-410f-b248-028f923cb281 (Rennie *et al.*, 2017i) |
| Spittle bugs (IS) | Annual | Count of each species/colour morph | https://doi.org/10.5285/aff433be-0869-4393-b765-9e6faad2a12b (Rennie *et al.*, 2018) |
| Baseline vegetation (VB) | One-off survey | Species presence | https://doi.org/10.5285/a7b49ac1-24f5-406e-ac8f-3d05fb583e3b (Rennie *et al.*, 2016a) |
| Coarse-grain vegetation (VC) | Every nine years | Species presence | https://doi.org/10.5285/d349babc-329a-4d6e-9eca-92e630e1be3f (Rennie *et al.*, 2016b) |
| Woodland vegetation (VW) | Every nine years - diameter at breast height (dbh) recorded every three years | See table 6 | https://doi.org/10.5285/94aef007-634e-42db-bc52-9aae86adbd33 (Rennie *et al.*, 2017j) |
| Fine-grain vegetation (VF) | Every three years – some sites did it annually | Species presence | https://doi.org/10.5285/b98efec8-6de0-4e0c-85dc-fe4cdf01f086 (Rennie *et al.*, 2017k) |
| Frog (BF) | Annual | See table 7 | https://doi.org/10.5285/4d8c7dd9-8248-46ca-b988-c1fc38e51581 (Rennie *et al.*, 2017l) |



| Breeding Bird Survey (BB) | Twice a year | Count of each species observed | https://doi.org/10.5285/5886c3ba-1fa5-49c0-8da8-40e69a10d2b5 (Rennie *et al*., 2017m) |
|---|---|---|---|
| Common Bird Census (CBC) | Annual (variable date ranges for sites) | Count of each species observed and/or nests observed | https://doi.org/10.5285/8582a02c-b28c-45d2-afa1-c1e85fba023d (Rennie *et al*., 2017n) |
| Bat (BA) | Four times a year | • Count of each species observed<br>• Behaviour | https://doi.org/10.5285/2588ee91-6cbd-4888-86fc-81858d1bf085 (Rennie *et al*., 2017o) |
| Rabbit and deer (BU) | Twice a year | Count of the dropping of each species | https://doi.org/10.5285/0be0aed3-f205-4f1f-a65d-84f8cfd8d50f (Rennie *et al*., 2017p) |

**Table 3: Meteorological variables**

| Name in Dataset | Description | Units |
|---|---|---|
| ALBGRD | Albedo Ground (average) | $Wm^{-2}$ |
| ALBSKY | Albedo Sky (average) | $Wm^{-2}$ |
| DRYTMP | Dry bulb temperature (average) | °C |
| DRTYMP_RH | Dry bulb temperature within the relative humidity sensor (average) | °C |
| NETRAD | Net Radiation (average) | $Wm^{-2}$ |
| RAIN | Rainfall (total) | mm |
| RH | Relative humidity (average) | % |
| SOLAR | Solar Radiation (average) | $Wm^{-2}$ |
| STMP10 | Soil temperature at 10 cm (average) | °C |
| STMP30 | Soil temperature at 30 cm (average) | °C |
| SURWET | Surface wetness (number of minutes in the hour that surface is wet) | minutes |
| SWATER | Soil moisture – gypsum block (average) | bar |
| SWATER_T | Soil moisture – theta probe at 20 cm (average) | % |
| SWATER_T10 | Soil moisture – theta probe at 10 cm (average) | % |
| SWATER_VWC | Soil moisture – volumetric water content at 20 cm (average) | $m^3/m^3$ |
| WDIR | Wind direction (average) | degrees |
| WETTMP | Wet bulb temperature (average) | °C |
| WSPEED | Wind speed (average) | $ms^{-1}$ |

**Table 4: Atmospheric Chemistry variables**





| Name in Dataset | Description | Units |
|---|---|---|
| WEIGHTNO2 | Weight of NO2 on the mesh | micrograms |
| NO2 | NO2 concentration | micrograms/m3 |
| NO2PPB | NO2 concentration | ppb |
| TDIFF | Exposure time | minutes |
| Q1-n | Quality code (see section 4) | integer |

**Table 5: Chemical and associated variables (Precipitation chemistry, soil solution, surface water chemistry)**

| Name in Dataset | Description | Units |
|---|---|---|
| ALKY | Alkalinity | mg/l |
| ALUMINIUM | Aluminium | mg/l |
| CALCIUM | Calcium | mg/l |
| CHLORIDE | Chloride | mg/l |
| COLOUR | Absorbance at 436nM | nM |
| CONDY | Conductivity | µS/cm |
| DOC | Dissolved organic carbon | mg/l |
| IRON | Iron | mg/l |
| MAGNESIUM | Magnesium | mg/l |
| NH4N | Ammonium | mg/l |
| NO3N | Nitrate nitrogen | mg/l |
| PH | pH | pH scale 1-14 |
| PHAQCS | Aquacheck system pH stirred | pH scale 1-14 |
| PHAQCU | Aquacheck system pH unstirred | pH scale 1-14 |
| PO4P | Phosphate phosphorus | mg/l |
| POTASSIUM | Potassium | mg/l |
| SO4S | Sulphate sulphur | mg/l |
| SODIUM | Sodium | mg/l |
| TOTALN | Total nitrogen | mg/l |
| TOTALP | Total dissolved phosphorus | mg/l |
| VOLUME | Volume of sample collected (Precipitation and soil solution chemistry datasets only) | ml |
| VACUUM | Residual vacuum at time of sampling | bar |





| | | |
|---|---|---|
| | (Soil solution chemistry dataset only) | |
| STAGE | Stage reading of water level (Surface water chemistry dataset only) | mm |

**Table 6: Woodland Vegetation variables**

| Name in dataset | DESCRIPTION | UNITS |
|---|---|---|
| A | Species recorded as sapling | species code |
| C | Species recorded as canopy dominant | species code |
| DIAMETER | Diameter at breast height (dbh) | cm |
| DISTANCE | Distance of stem from centre of random cell | m |
| E | Species recorded as seedling | species code |
| H | Species recorded as shrub layer | species code |
| HEIGHT | Height | m |
| I | Species recorded as intermediate | species code |
| NUM_STEMS | Number of stems | count |
| S | Species recorded as subdominant | species code |
| SEEDLING | Species recorded in seedling survey of cell | species code |
| U | Species recorded as suppressed | species code |
| Q1-n | Quality code (see section 4) | integer |

5  **Table 7: Frog variables**

| Name in dataset | Description | Units |
|---|---|---|
| ALKY | Alkalinity | mg/l |
| ALUMINIUM | Aluminium | mg/l |
| CALCIUM | Calcium | mg/l |
| CHLORIDE | Chloride | mg/l |
| CONDY | Conductivity | µS/cm |
| COLOUR | Absorbance at 436nM | nM |
| CONGDATE | Date frogs first seen congregating | date |
| DEPTH | Depth at centre of spawning area | cm |
| DOC | Dissolved organic carbon | mg/l |
| HATCHDATE | Date of first hatching observed | date |
| IRON | Iron | mg/l |





| LEAVEDATE | Date frogs first seen leaving | date |
| --- | --- | --- |
| MAGNESIUM | Magnesium | mg/l |
| MAXTMP | Maximum temperature | °C |
| MINTMP | Minimum temperature | °C |
| NEWMASS | Number of new spawn masses | count |
| NH4N | Ammonium | mg/l |
| NO3N | Nitrate nitrogen | mg/l |
| PERCDEAD | Percentage dead or diseased eggs | % |
| PH | pH from water sample processed in laboratory | pH scale 1-14 |
| PH1 | First pH reading from daily sample | pH scale 1-14 |
| PH2 | Second pH reading from daily sample | pH scale 1-14 |
| PH3 | Third pH reading from daily sample | pH scale 1-14 |
| PHAQCS | Aquacheck system pH stirred | pH scale 1-14 |
| PHAQCU | Aquacheck system pH unstirred | pH scale 1-14 |
| PO4P | Phosphate phosphorus | mg/l |
| POTASSIUM | Potassium | mg/l |
| SO4S | Sulphate sulphur | mg/l |
| SODIUM | Sodium | mg/l |
| SPAWNDATE | Date of first spawning observed | date |
| SURFAREA | Total surface area covered by spawn | $m^2$ |
| STAGE | Stage reading of water level | mm |
| TOTALN | Total nitrogen | mg/l |
| TOTALP | Total dissolved phosphorus | mg/l |
| VACUUM | Residual vacuum at time of sampling | bar |
| VOLUME | Volume of sample collected | ml |
| Q1-n | Quality code (see section 4) | integer |