# Peer review of "The UK Environmental Change Network datasets - integrated and colocated data for long-term environmental research (1993-2015)"

_Earth System Science Data, 2019_

## Short Comment (SC1) · 26 Jul 2019

General comments This is a interesting dataset, in particular since standardised observations protocols have been used across a large geographical and multidisciplinary measurement network. More general questions that arises when reading the document include: 1. Given the time series approach of the dataset, has it been considered to homogenise the time series? That would increase the time series approach of the dataset. A comment on this would make sense since part of the justification for the combined dataset is to analyse the temporal evolution of features. 2. Documenting data using discipline specific standards (e.g. NetCDF Climate and Forecast conven-

tion or Global Biodiversity Information Facility standards) simplifies reuse since data are documented using controlled vocabularies to describe variables, their units, cell methods etc. Has application of standardised documentation been considered for the dataset and if not why?

Specific comments Page 4, section 2.1: The text refers to full operating procedures in a separate document. Given the nature of some of the meteorological parameters that require more maintenance than standard meteorological observations (e.g. surface irradiance) it would be beneficial to have some more explanation of how these parameters are handled in this document. Page 4, section 2.1: It would also be natural to describe the sampling frequency in this document for consistency with other sections below although it is acknowledged that meteorological observations are slightly more complex to describe in a simple manner than the other observations due to the number of parameters. Page 4, section 2.1: The AWS are located according to the handbook of 1982, but how are stations constructed, at which levels are sensors located etc and how are sensors maintained. Is that following any larger scale framework observation protocol like WMO? Page 5, section 2.5: It would be beneficial to include frequency of dip samples similar to how this is indicated in section 2.4. Page 8, section 2.16: It is commented that the methodology for bird observations changed during the time series, but it is not commented on how these two approaches compare and how that affects potential analysis of the time series. Page 8, section 2.18: Reference for the Bats and Habitats survey methodology of the Joint Nature Conservation Committee is missing. Page 11, section 3.1: The text refers to the AWSNO field but doesn't explain it in more detail (which type of information is provided, binary change – no change or id numbers, or something else). More information would be beneficial since this field is commented in the document, although it is acknowledged that full details are in the reference (which probably should be repeated here). The presence of the AWSNO and the text provided caused the general question on homogenisation of the time series. There is also a comment that the dataset is so large, but what does that mean in this context? Numbers would be good. Page 11, section 3.3: Again a comment on size of

the dataset, but no explanation or justification is provided. Page 11, sections 3.9 an 3.5 (wrong numbers): Did you consider using GBIF standards for these datasets? Page 12, section 4.2: It would be beneficial with some more information on the templates developed. It is not clear whether the templates were developed for simplifying the data entry process, quality assure the data or the entry process? How many templates were developed etc? This is an interesting element for reuse of the data and in particular if human errors are captured. Page 12, section 4.3: The relation between sections 4.2 and 4.3 could be further explained. Is data verification done in the templates mentioned in section 4.2 as well as in a separate step? Page 12, section 4.3: Where are ranges for the ECN variables defined and where is the process leading up to these ranges documented? It is also commented that data out of range were treated in 3 different ways. On the second bullet point, what was the consequence for the data? Were data corrected and versioned? Page 12, section 4.3: Please consider referring to section 4.4 for explanation of quality flags. Page 13, section 4.5: Again some further description of the processes around the meteorological data would be good, in particular irradiance which has issues concerning ventilation etc. And where sensors or AWS were run in parallel for periods, did they compare well? Page 13, section 5: Some more discussion on the temporal scales the dataset can be used for concerning non homogenised data would be beneficial. This would of course also depend on the types os analysis done and e.g. how sensitive the biosphere is to climate parameters.

Technical corrections Page 11, section 3.9: Numbering must be wrong.

---

## Referee Comment (RC2) · Johannes Peterseil (Referee) · 5 Nov 2019

[referee-annotated manuscript omitted]

---

## Author Response (AR1)

We thank the Øystein Godoy and Johannes Peterseil for their thoughtful and helpful comments. Our responses (coloured red) are added next to the relevant comment – these have also been highlighted with comments in the revised manuscript.

**Referee 1 - Oystein Godoy**

"General comments This is a interesting dataset, in particular since standardised observations protocols have been used across a large geographical and multidisciplinary measurement network. More general questions that arises when reading the document include: 1. Given the time series approach of the dataset, has it been considered to homogenise the time series? That
10   would increase the time series approach of the dataset. A comment on this would make sense since part of the justification for the combined dataset is to analyse the temporal evolution of features. 2. Documenting data using discipline specific standards (e.g. NetCDF Climate and Forecast convention or Global Biodiversity Information Facility standards) simplifies reuse since data are documented using controlled vocabularies to describe variables, their units, cell methods etc. Has application of standardised documentation been considered for the dataset and if not why?"

15   **Referee 1 - Comment 1 -** ECN's policy has been to provide data as they are collected – which we have quality checked - and allow users to analyse the data in the way that suits them. The ECN protocols cover a number of domains (meteorology, biogeochemistry, biodiversity etc) and we do not have the resources to fine-tune the datasets in a way that a domain-specific data centre could do. We will however seek opportunities in the future to work with domain experts to ensure our data are useful to as many users as possible.

20   We provided information on where to find information about the standard terms used in the supporting documentation in section 3.5. We primarily use the EnvThes controlled vocabulary – which was developed as part of our involvement in the LTER-Europe community - as the basis for the semantic harmonisation of data with our European and International partners. We have now expanded this section in the text to include this information.

Since our data cover a number of domains, and our users tend to use several datasets together (e.g. to explore the effects of
25   weather on species abundance), we made the choice to provide CSV files to users so they can use them together easily - rather than use a variety of discipline-specific documentation standards.

"Specific comments Page 4, section 2.1: The text refers to full operating procedures in a separate document. Given the nature of some of the meteorological parameters that require more maintenance than standard meteorological observations (e.g. surface irradiance) it would be beneficial to have some more explanation of how these parameters are handled in this
30   document."

**Referee 1 – Comment 2 -** The ECN AWS' had regular, professional maintenance by external contractors on an annual or bi-annual basis. We've added this information to the manuscript.

"Page 4, section 2.1: It would also be natural to describe the sampling frequency in this document for consistency with other sections below although it is acknowledged that meteorological observations are slightly more complex to describe in a simple manner than the other observations due to the number of parameters."

**Referee 1 – Comment 3** - The frequency of data collection for all the ECN protocols is provided in table 2 but we have also now made this clearer in the manuscript.

"Page 4, section 2.1: The AWS are located according to the handbook of 1982, but how are stations constructed, at which levels are sensors located etc and how are sensors maintained. Is that following any larger scale framework observation protocol like WMO?"

**Referee 1 – Comment 4** - We have added a diagram to the manuscript to show the layout of the meteorological enclosure and also added text which describes the height that the sensors are installed at and maintenance information. As noted in the manuscript the AWS was sited in accordance with British Meteorological Office requirements.

"Page 5, section 2.5: It would be beneficial to include frequency of dip samples similar to how this is indicated in section 2.4."

**Referee 1 – Comment 5** - The frequency of data collection for all the ECN protocols is provided in table 2 but we have also now made this clearer in the manuscript.

"Page 8, section 2.16: It is commented that the methodology for bird observations changed during the time series, but it is not commented on how these two approaches compare and how that affects potential analysis of the time series."

**Referee 1 – Comment 6** - The methodologies of the two surveys are different so it is unfortunately not possible to create a single time series from both datasets. We have made this clearer in the manuscript.

"Page 8, section 2.18: Reference for the Bats and Habitats survey methodology of the Joint Nature Conservation Committee is missing."

**Referee 1 – Comment 7** - Thank you. We've added this reference to the text.

"Page 11, section 3.1: The text refers to the AWSNO field but doesn't explain it in more detail (which type of information is provided, binary change – no change or id numbers, or something else). More information would be beneficial since this field is commented in the document, although it is acknowledged that full details are in the reference (which probably should be repeated here). The presence of the AWSNO and the text provided caused the general question on homogenisation of the time series…..

**Referee 1 – Comment 8** - We have added some information on how AWSNO are assigned to the manuscript.

…..There is also a comment that the dataset is so large, but what does that mean in this context? Numbers would be good."

**Referee 1 – Comment 9** - We have provided information in section 3 to cover usage issues that our users have encountered in the past. Occasionally our users have been unable to open the data file – if they use a software package like Excel - because of the file size. We have therefore provided our normal advice in the manuscript to help users overcome this issue. This has been made clearer in the manuscript.

"Page 11, section 3.3: Again a comment on size of the dataset, but no explanation or justification is provided."

**Referee 1 – Comment 10** - See comment above re file size.

"Page 11, sections 3.9 an 3.5 (wrong numbers): Did you consider using GBIF standards for these datasets?"

**Referee 1 – Comment 11** - We provided information on where to find information about the standard terms used in the supporting documentation in section 3.5. As mentioned above, we primarily use the EnvThes controlled vocabulary for the semantic harmonisation of our data with our partners. We have now expanded this section in the text.

"Page 12, section 4.2: It would be beneficial with some more information on the templates developed. It is not clear whether the templates were developed for simplifying the data entry process, quality assure the data or the entry process? How many templates were developed etc? This is an interesting element for reuse of the data and in particular if human errors are captured."

**Referee 1 – Comment 12** - We have included a reference which provides more detailed information about the templates.

"Page 12, section 4.3: The relation between sections 4.2 and 4.3 could be further explained. Is data verification done in the templates mentioned in section 4.2 as well as in a separate step?"

**Referee 1 – Comment 13** - This verification is in addition to the checks made in the templates, this has been made clearer in the text.

"Page 12, section 4.3: Where are ranges for the ECN variables defined and where is the process leading up to these ranges documented? It is also commented that data out of range were treated in 3 different ways. On the second bullet point, what was the consequence for the data? Were data corrected and versioned?"

**Referee 1 – Comment 14** - The ranges are held in the database, this has been added to the manuscript.

As the second bullet says the data were corrected and the changes flagged in the database.

"Page 12, section 4.3: Please consider referring to section 4.4 for explanation of quality flags."

**Referee 1 – Comment 15** - Thank you. This has been added to the manuscript.

"Page 13, section 4.5: Again some further description of the processes around the meteorological data would be good, in particular irradiance which has issues concerning ventilation etc. And where sensors or AWS were run in parallel for periods, did they compare well?"

**Referee 1 – Comment 16** - Further information on this has been added to the manuscript.

"Page 13, section 5: Some more discussion on the temporal scales the dataset can be used for concerning non homogenised data would be beneficial. This would of course also depend on the types os analysis done and e.g. how sensitive the biosphere is to climate parameters."

**Referee 1 – Comment 17** - A number of references have been provided to demonstrate the breadth of research which uses ECN data. The Data Centre does not have the resources to do these type of analyses itself so users are recommended to review the highlighted literature to explore how the data can be used.

"Technical corrections Page 11, section 3.9: Numbering must be wrong."

**Referee 1 – Comment 18 -** Thank you – we've corrected the numbering.

**Referee 2 – Johannes Peterseil**

The data paper provides a good overview on the data provided from the UK ECN Network. Method and dataset descriptions are clear. I checked several (not all) of the DOI and links are working providing the described content.

Observation sites - reference to the documentation of the observaiton sites (if existing) on DEIMS-SDR should be provided to provide a further description of the sites to the user. This also should be added to table 1.

**Referee 2 – Comment 1 –** This information has been included in the introduction and added to table 1.

Method description - temporal scale of observation is provided in table 2 but should also be mentioned in the textual description of the method. This would be a benefit to the reader. In addtion table 2 should be mentioned in the '2. Methods'.

**Referee 2 – Comment 2 –** The frequency of data collection for all the ECN protocols is provided in table 2 but we have also now made this clearer in the manuscript.

**Comments provided in the supplement**

Page 2, line 20: reference to Mirtl et al. 2008 / Mirtl et al. 2019 STOTEN, refererring to the global network of LTER sites. ILTER and LTER Europe are important contributions to this effort.

**Referee 2 – Comment 3 –** The Sci. Tot. Env. reference has been added to the manuscript

Page 3, line 5: structure?

**Referee 2 – Comment 4 –** We think that 'system' would be more appropriate

Page 4, line 6: remove ":" as the following are sub-chapters

**Referee 2 – Comment 5 –** this has been altered in the manuscript

Page 4, line 7: General comment: for some of the datasets the temporal resolution is mentioned in the textual description, but not for all. This should bbe done for all, or mentioning in the paragraph above, that temporal resolution can be found in Table 2.

**Referee 2 – Comment 6 -** The frequency of data collection for all the ECN protocols is provided in table 2 but we have also now made this clearer in the manuscript.

Page 4, line 16: maybe a better term "control measure of blank tubes ..."

**Referee 2 – Comment 7 –** this has been altered in the manuscript

Page 5, line 22: see comment above. would be good to mention temporal resolution of the sample collection in order to allow to assess the data use.

**Referee 2 – Comment 8 -** The frequency of data collection for all the ECN protocols is provided in table 2 but we have also now made this clearer in the manuscript.

Page 7, line 16: was there a repetition of the baseline vegetation survey? should be mentioned also in the text.

**Referee 2 – Comment 9 –** As stated in the manuscript, this was a one-off survey.

Page 7, line 29: how was the number of random plots selected defined? was this by the share of ecosystem types? should be mentioned with one sentence.

**Referee 2 – Comment 10 –** Additional information has been provided to make this clearer in the manuscript.

Page 10, line 15: this sentence should be placed into the introduction section at the end.

**Referee 2 – Comment 11 –** Reference to table 2 has been included in the introduction

Page 10, line 16: would be good to start with a summary on the structure of the "data package" containing the data as well as the supporting documentaiton. The full package is referenced by a DOI. please consider to reorder the paragraphs a bit.

**Referee 2 – Comment 12 –** This section has been reordered

Page 10, line 16: as for meteorology it is specifically mentioned that the datasets were split into yearly time slices, the general rule for providing the data should be mentioned. I assume that in general the whole period is provided in one dataset file, correct?

**Referee 2 – Comment 13 –** Yes this is correct, this has been clarified in the manuscript.

Page 10, line 24: put this at the beginning of the paragraph "datasets"

**Referee 2 – Comment 14 -** This section has been reordered

Page 11, line 3: replace ":" by "."

**Referee 2 – Comment 15 –** This has been altered.

Page 11, line 5: see comment above

**Referee 2 – Comment 16 –** This has been clarified in the manuscript

Page 11, line 7: should also be mentioned in the method section on meteorology

**Referee 2 – Comment 17 –** This has been added to the manuscript

Page 11, line 9: is this mentioned in the "supporting documents" or in the metadata?

**Referee 2 – Comment 18 –** Yes, this information is included in the supporting documentation

Page 11, line 28: please put a short reference to the speices lists used - as a summary of the supporting documentation.

**Referee 2 – Comment 19 –** This has been added as a new table (table 8).

Page 11, line 29: is there also an online ressource the user can access?

**Referee 2 – Comment 20 –** We do not make our species list available online but we primarily use the EnvThes controlled vocabulary for the semantic harmonisation of our data with our partners. We have now expanded this section in the text.

Page 12, line 9: replace ":" by "a number of steps."

**Referee 2 – Comment 21 -** This has been altered.

Page 12, line 17: is there a reference describing this data processing steps? online reference to online description of the data reporting templates available?

**Referee 2 – Comment 22 –** A reference has been added.

Page 14, line 24: this could be already mentioned in the introduction. please consider to move this paragraph into the introduction.

**Referee 2 – Comment 23 -** This section seeks to show the context in which the ECN datasets are collected and why they are useful to researchers. Therefore we think it is appropriate to include this information in this section. However we have included some additional information about this in the introduction.

Page 14, line 30: could be moved after the chapter "4 . datasets" to put datasets relevant information together.

**Referee 2 – Comment 24 –** Our reading of the ESSD manuscript preparation guidelines for authors is that the Data Availability section should appear just before the conclusions. We are happy to re-order if we have misunderstood.

Page 15, line 3: please mention, that registration is necessary to download the data!

**Referee 2 – Comment 25 –** This has been added to the manuscript

Page 15, line 7: how will this be done? will there be an updated full period datasets or additional yearly data slices? if possible, please add information here.

**Referee 2 – Comment 26 -** This has been added to the manuscript

Page 25, table 1: In order allow a better reference and access to online documentation of the observation sites reference to the documentation on DEIMS-SDR (https://deims.org/) is recommended. Please add the deims.id for the sites in the table.

**Referee 2 – Comment 27 –** The DEIMS IDs for the ECN sites has been added to table 1.

Page 27, table 2: remove listing points

**Referee 2 – Comment 28 –** This has been done

[revised manuscript text omitted]

* * *
**Commented [RSC10]:** Response to Referee 1 – Comment 3 and Comment 5
Response to Referee 2 – Comment 2, Comment 6 and Comment 8

**Commented [RSC11]:** Altered as a result of Referee 1 – Comment 4

**Commented [RSC12]:** Response to Referee 2 – Comment 7

**Commented [RSC13]:** Response to Referee 1 – Comment 3 and Comment 5
Response to Referee 2 – Comment 2, Comment 6 and Comment 8

**Commented [RSC14]:** Altered as a result of Referee 1 – Comment 4

**Commented [RSC15]:** We added to clarify which organisations we are referring to

**Commented [RSC16]:** Response to Referee 1 – Comment 3 and Comment 5
Response to Referee 2 – Comment 2, Comment 6 and Comment 8

**2.4 Soil Solution Chemistry**

Water was collected from soils via suction lysimeters at the majority of ECN terrestrial sites. The lysimeters were installed at two depths within a 10m by 10m plot on the edge of, but outside, the TSS. Six samplers were installed in the A horizon and six others at the base of the B horizon (or at 10cm and 50cm if these soil horizons did not exist), ideally on a downslope to avoid debris from soil disturbance. Samplers were emptied and the water volumes collected on the same day each fortnight. One week after sample collection, the samplers were evacuated to 0.5 bar (or 0.7 bar for sites where insufficient soil solution could be collected), so the water only accumulated over the second week of the fortnightly period. The chemistry of the water collected was analysed by the analytical labs associated with each site. At some sites, particularly in drier months, the volume of water collected may have been very small; in these cases, the samples were discarded or, if possible, combined (only samples from the same horizon were combined) for analysis (see section 3.2. for details on how this is recorded in the dataset). The samples were collected fortnightly and the variables recorded are listed in table 5. Full operating procedures are provided in the protocol document (Adamson, 1996b) which is included in the supporting documentation provided alongside the data download (called SS.pdf). Operating procedures for handling water samples (Adamson, 1996a) and analytical guidelines (Rowland, 1996) are also provided in the supporting information (called WH.pdf and WAG.pdf).

**2.5 Surface Water Chemistry**

Dip samples from rivers and streams were collected. This was only done at sites where flowing water was present. Samples were taken at a representative location above a weir; some sites collect samples at multiple locations on the site (indicated by the location code in the dataset). The collecting bottle is rinsed in river water and a 250ml sample of river water taken. The samples were collected weekly and the variables recorded are listed in table 5. Full operating procedures are provided in the protocol document (Johnson and Burt, 1996a) which is included in the supporting documentation provided alongside the data download (called WC.pdf). Operating procedures for handling water samples (Adamson, 1996a) and analytical guidelines (Rowland, 1996) are also provided in the supporting information (called WH.pdf and WAG.pdf)

**2.6 Surface Water Discharge**

Hydrological data from rivers and streams were collected by logger at sites with a river or stream. Recording of river stage was by a permanently installed weir, the design of which was determined by the conditions at the site. Data were recorded by a logger. The data are 15-minute averages calculated from ten second samplings of stage height and tThe variables recorded are listed in table 2. Full operating procedures are provided in the protocol document (Johnson and Burt, 1996b) which is included in the supporting documentation provided alongside the data download (called WD.pdf).

**Commented [RSC17]:** We added this to aid the reader

**Commented [RSC18]:** Response to Referee 1 – Comment 3 and Comment 5
Response to Referee 2 – Comment 2, Comment 6 and Comment 8

**Commented [RSC19]:** Response to Referee 1 – Comment 3 and Comment 5
Response to Referee 2 – Comment 2, Comment 6 and Comment 8

**Commented [RSC20]:** Response to Referee 1 – Comment 3 and Comment 5
Response to Referee 2 – Comment 2, Comment 6 and Comment 8

[revised manuscript text omitted]

**Commented [RSC22]:** Response to Referee 2 – comment 10

**Commented [RSC23]:** Response to Referee 2 – Comment 10

**Commented [RSC24]:** Response to Referee 1 – Comment 3 and Comment 5
Response to Referee 2 – Comment 2, Comment 6 and Comment 8

[revised manuscript text omitted]

Commented [RSC36]: Response to Referee 1 – Comment 18

Commented [RSC37]: We have added this reference to provide more information to readers

Commented [RSC38]: Response to Referee 1 – Comment 1 and Comment 11
Response to Referee 2 – Comment 20

Commented [RSC39]: Response to Referee 2 – Comment 19

Commented [RSC40]: Response to Referee 2 Comment 21

across sites and over time. This documentation includes rules for handling missing values and data quality information. To aid site managers a bespoke set of data entry templates were developed for each protocol, using MS Access, to improve data handling efficiency (Rennie, 2016). These templates incorporate quality checking procedures, and help to ensure that quality checked, standardised and formatted data were submitted by site managers. The design of the templates takes into account ease of use, with the main emphasis being on minimising error. This type of data entry software is particularly useful where numeric coding systems for species are in use; numbers are less memorable and mistakes in one digit of a code can produce serious errors. For example, the software uses drop-down lists of codes (which are dynamically linked with a list of the species names) so that the codes can be cross-checked against the species name to ensure that the correct code is chosen.

**4.3 Data Verification**

In addition to the checks made in the templates, Sstandard verification procedures were applied to all data before import into the database. The procedures performed numeric range checks (i.e. checking if a value falls within a specified range), categorical checks (e.g. checking that a species code appears on the standard code list), formatting (i.e. that the dataset conforms to the specified data format) and logical integrity checks (i.e. checking the data make sense e.g. that the dates in one dataset match those in a related dataset). Appropriate range settings for ECN variables were selected following discussion with specialists in each field. These ranges are held in a table in the database and the data are checked against this before being committed to the database. Where values fell outside these ranges, a cautious approach was adopted towards discarding data on the principle that apparent errors could be valid outliers. Data values identified by validation software as 'out of range' were treated in one of three ways:

- where values were clearly meaningless due to a known cause, (e.g. an instrumentation fault, and could not be back-corrected), the data were discarded and database fields set to null (no data), and quality flags added to the database;
- where values were clearly in error, or out of range due to known calibration errors and could be back-corrected, the data were corrected. These changes were flagged in the database;
- where there was no straightforward explanation for outliers, the data were stored in the database, accompanied by quality flags (see section 4.4).

**4.4 Quality Flagging**

The ECN site managers assigned quality codes to indicate factors that may affect the quality of the data being collected, including deviations from the protocol, faulty instrumentation and common problems. They picked these from a standard list of ECN quality codes and these quality codes are included in the data download and an explanation for the codes provided in the supporting documentation. Site managers could pick as many quality codes as were applicable. Occasionally, an unusual event took place that was not covered by these codes. In that case, the site manager attached text explaining the circumstances. This is indicated by a quality code '999' in the data download. This quality text is available in a file called ECN_***_qtext.csv (where *** is the measurement code; see table 2) which is provided in the supporting documentation.

Commented [RSC41]: Response to Referee 1 – Comment 12
Response to Referee 2 – Comment 22

Commented [RSC42]: Response to Referee 1 – Comment 13

Commented [RSC43]: Response to Referee 1 – Comment 14

Commented [RSC44]: Response to Referee 1 – Comment 15

**4.5 Quality Assessment Exercises**

Samples were kept where possible (e.g. archived invertebrate samples) meaning the accuracy of identification can been assessed at a later date if necessary. Occasionally, quality assessment exercises have been run by appropriate experts to check, for example, consistency in species identification across sites (Scott and Hallam, 2003). The quality of more ephemeral measurements such as meteorology or water quality can only be similarly assessed by running duplicate or parallel systems. Duplicate systems are expensive, and in practice assessment normally involved regular checks for instrument drift and recorder error. Where possible, when new instrumentation or methods needed to be introduced, new and old systems were run in parallel to assess their relationship. This is assessed by the individual site manager who must satisfy themselves that the new systems compare well before proceeding with the switchover.

Commented [RSC45]: Response to Referee 1 – Comment 16

**5. ECN datasets in context**

ECN is nationally unique with its focus on high frequency and co-located measurements. It provides a rare opportunity to link pressures and responses to investigate relationships between environmental variables and explore environmental change over significant time scales. The data included within these datasets have been the focus of a number of peer-reviewed scientific publications over the past 20 years. For example, linking meteorological with invertebrate species data to explore the impact of drought (Morecroft *et al*., 2002); exploring trends in the physical and biological environment (Morecroft *et al.*, 2009); determining that hydrochloric acid deposition was a driver of UK soil acidification (Evans, *et al*., 2011); and to investigate declines in carabid beetle biodiversity (Brooks *et al*., 2012). Many of the datasets were incorporated in papers forming a journal special issue marking the first 20 years of ECN (Sier and Monteith (*eds.*), 2016b). This special issue demonstrates how effective the datasets are in assessing and interpreting environmental change, covering a breadth of topics such as trends in weather and atmospheric deposition (Monteith, *et al*., 2016); trends in dissolved organic carbon (Sawicka *et al.*, 2016; Moody *et al.*, 2016); various aspect of change in UK plant communities (Rose *et al.*, 2016; Morecroft *et al.*, 2016; Pallett *et al.*, 2016; Milligan *et al*., 2016); ecosystem services (Dick *et al.*, 2016); carabid beetle communities (Eyre *et al.*, 2016; Pozsgai *et al*., 2016); the use of digital imaging to assess vegetation cover (Baxendale *et al.*, 2016); and the response of Lepidoptera communities to warming (Martay *et al.*, 2016). A full catalogue of the peer-reviewed papers that have used ECN data are available on the website (ECN Publications Catalogue, 2019).

ECN sites cover a wide range of UK habitats but, given their focus on high frequency data, are costly to run and are relatively few in number. The representativeness of ECN sites was compared to data obtained by the UK Countryside Survey (CS - Countryside Survey, 2019). The survey is based on a stratified random sample of 1 km squares from the intersections of a regular 15 km grid superimposed on the rural areas of Great Britain. Analysis revealed that ECN sites effectively span the range of values for both temperature and rainfall and cover a similar range of vegetation types to the CS with the exception of arable, a land use category not assessed at ECN sites although present on several sites (Dick *et al*., 2011).

ECN sites contribute to a number of national monitoring programmes e.g. Rothamsted Insect Survey (Rothamsted Insect Survey, 2019), Countryside Survey (Countryside Survey, 2019), the UK Butterfly Monitoring Scheme (UKBMS, 2019), the Breeding Bird Survey (BBS, 2019), the United Kingdom Eutrophying and Acidifying Network (UKEAP, 2019) and the Cosmic-ray Soil Moisture Monitoring Network (COSMOS-UK, 2019). ECN's focus on multidisciplinary, co-located

5    measurements can help integrate these networks and provides temporal scale context for observations made by these networks, for example by providing information on year to year variation in vegetation communities to help inform how CS data can be influenced by weather variability (Scott *et al.*, 2010).

ECN is formally recognised as the UK's contribution to a global system of long-term, integrated environmental research networks and is a member of LTER-Europe (the European Long-Term Ecosystem Research Network – Mirtl, 2010) and ILTER

10    (International Long Term Ecological Research – Kim, 2006). Individual ECN sites are also involved in other international networks, including INTERACT (International Network for Terrestrial Research and Monitoring in the Arctic - INTERACT, 2019), GLORIA (Global Observation Research Initiative in Alpine Environments - GLORIA, 2019), ICP Forest Level II (ICP Forests, 2019) and FLUXNET (FLUXNET, 2019).

**6. Data Availability**

15    Provision of easy access to data has always been central to ECN's strategy to provide a resource for environmental research, policy purposes and public information. The ECN datasets are hosted by the NERC Environmental Information Data Centre (EIDC, 2019) managed by the Centre for Ecology and Hydrology (CEH). The EIDC manages nationally important terrestrial and freshwater science datasets and is a CoreTrustSeal accredited data repository. EIDC has a registration system - users need a free account to download data. The ECN datasets can be discovered and downloaded through the EIDC's data catalogue

20    (the Environmental Information Platform (EIP)). The datasets are listed in table 2, together with their citation information. They should be cited for every use using the information provided (Rennie *et al*., 2016a,b; Rennie *et al*., 2017a-p; Rennie *et al*., 2018).

The ECN datasets are available under the Open Government Licence (Open Government Licence, 2019) and they are available as comma-separated files. Temporal extensions, provided as additional time slices, to the datasets will be created as additional

25    furtheryears of data become available.

**Commented [RSC46]:** The Data Centre recently received this accreditation so we have added this information to help users feel confident that the data are from a trustworthy source.

**Commented [RSC47]:** Response to Referee 2 – Comment 26

[revised manuscript text omitted]

Commented [RSC52]: We have added this reference to provide more information to readers

Commented [RSC53]: Response to Referee 1 – Comment 7

Commented [RSC54]: Added in response to Referee 2, Comment 1

Woiwod, I.P. and Coulson, J.C.: Ground predators, in: The United Kingdom Environmental Change Network: Protocols for standard measurements at terrestrial sites, edited by: Sykes, J.M. and Lane, A.M.J., The Stationery Office (London), 118-121, 1996.

**Figure 1: Locations of the ECN terrestrial sites**

[Figure]

**Figure 2: ECN Meteorological Enclosure (MA = Automatic Weather station; PC = Precipitation Chemistry; AN = Atmospheric Nitrogen; MM = Manual meteorology)**

[Figure]

**Table 1: ECN terrestrial sites**

| Site (ECN Site code) | Site Description (links to the ECN website and DEIMS-SDR) | Location | Altitudinal Range (m above sea level) | Area (ha) | Site Type |
|---|---|---|---|---|---|
| Alice Holt (T09) | http://data.ecn.ac.uk/sites/ecnsites.asp?site=T09 | 51° 9'16.46"N | 110-125 | 850 | Woodland |

Commented [RSC55]: Response to Referee 1 – Comment 4

Commented [RSC56]: Response to Referee 2 – Comment 1 and Comment 27

Formatted Table

Field Code Changed

[revised manuscript text omitted]